# Barriers to Human Papillomavirus Vaccination Initiation and Completion among Adults Aged 18–26 Years in a Large Healthcare System

**DOI:** 10.3390/cancers15174243

**Published:** 2023-08-24

**Authors:** Lucien Khalil, Elena Russo, Kripa Venkatakrishnan, Angela L. Mazul, Jose P. Zevallos

**Affiliations:** 1Department of Otolaryngology, Head and Neck Surgery, University of Pittsburgh Medical Center, Pittsburgh, PA 15213, USA; khalill3@upmc.edu (L.K.); angela.mazul@pitt.edu (A.L.M.); 2Department of Biomedical Sciences, Humanitas University, Via Rita Levi Montalcini 4, Pieve Emanuele, 20090 Milan, Italy; elena.russo@humanitas.it; 3Otorhinolaryngology Unit, IRCCS Humanitas Research Hospital, Via Manzoni 56, Rozzano, 20089 Milan, Italy; 4Department of Clinical Analytics, University of Pittsburgh Medical Center, Pittsburgh, PA 15213, USA; venkatakrishnank3@upmc.edu

**Keywords:** HPV vaccination, cancer prevention, vaccination prevalence, sociodemographic factors

## Abstract

**Simple Summary:**

This study investigated HPV vaccination rates and associated factors among 265,554 patients aged 18–26 within a large healthcare system. Alarmingly, only 30% had completed the vaccine series, far below national averages and targets. Significant disparities were found, with males less likely to be vaccinated than females and Black Americans less likely to complete vaccination despite higher initiation rates. Many patients remained unvaccinated despite recent primary care visits, highlighting missed opportunities. Low uptake, particularly among males, and disparities emphasize the urgent need for targeted interventions to improve provider communication and community education around HPV risks and vaccine benefits to increase vaccination coverage in this population.

**Abstract:**

Human papillomavirus (HPV) is a common sexually transmitted infection, with over 40% prevalence in the US. Oropharyngeal cancers (OPCs) driven by high-risk HPV are increasing (up to 90%), with HPV vaccination being the only prevention available. The aim of this study was to investigate HPV vaccination among patients aged between 18 and 26 years old with at least one encounter at a large healthcare system and identify sociodemographic factors associated with vaccine initiation and completion. A cross-sectional retrospective study was conducted between 2018 and 2021, including 265,554 patients identified from the Clinical Data Warehouse. HPV vaccination status by age, sex, race/ethnicity, insurance type, primary care (PCP) visits in the past year, alcohol, tobacco, illicit drug use, and age at vaccination was examined. Overall, 33.6% of females and 25.4% of males have completed the HPV vaccine. Black Americans were 35% more likely to initiate the vaccine than White Americans but were less likely to complete the entire course. Overall, HPV vaccination prevalence was far below the Health People 2030 goal of 80%, especially in young males. This low rate is troubling, since many patients had a PCP visit and remained unvaccinated, which serves as a missed opportunity for vaccination.

## 1. Introduction

Human papillomavirus (HPV) is the most common sexually transmitted infection in the United States (US), with over 40% of people infected [1]. The symptoms of HPV infection vary depending on the site and subtype of the virus, ranging from a benign wart to an HPV-associated malignancy. HPV is a known cause of certain types of cancer like cervical cancer and vaginal, vulvar, anal, penile, and oropharyngeal cancers. According to the Centers for Disease Control and Prevention (CDC), more than 45,000 annual cases of HPV-related malignancies are diagnosed yearly. While primary and secondary prevention strategies have significantly reduced the incidence of HPV-associated cervical cancer [2], the rates of other HPV-associated cancers, especially oropharyngeal cancer, which is now the most common cancer caused by HPV [3], have been increasing in recent years [4]. Moreover, it has been suggested that women who have undergone prior treatment for cervical precancerous lesions could face an elevated risk of developing subsequent HPV-associated malignancies [5]. Nonetheless, there is limited understanding regarding the extent of risk associated with these secondary tumors, as well as the efficacy of HPV vaccination in individuals who have already received treatment.

HPV vaccination is highly effective and recommended for boys and girls between 11 and 12 years old and can be started at nine years old, with catch-up vaccines at the age of 26. Vaccination is the only primary prevention available for oropharyngeal cancer. While HPV vaccination is not recommended after 26 years of age, some patients between 27 and 45 years old could benefit depending on whether they were vaccinated adequately when younger after discussing it with their physician [6]. HPV vaccination can provide benefits even in cases of an existing infection. Remarkably, it has shown the capability to diminish the risk of a persistent HPV infection and aid in the regression of cervical precancerous lesions [7]. Furthermore, it appears to play a role in decreasing the prevalence of high-risk HPV genotype infections among individuals who have previously tested positive for them [8]. In 2020, 54.5% of adolescents aged between 13 and 15 years old had received at least two or three vaccine doses [9,10], far from the Healthy People 2030 goal of 80%.

In this study, an estimate is provided of HPV vaccination uptake among eligible adults aged between 18 and 26 years old within the University of Pittsburgh Medical Center (UPMC) hospital system, which is a large private healthcare system operating multiple hospitals. The aim was to identify which sociodemographic factors are associated with low vaccination initiation and completion in this population to reduce the burden of diseases related to HPV in vulnerable populations.

## 2. Materials and Methods

### 2.1. Study Design

Data were collected from Clinical Data Warehouse, an extensive database that stores healthcare data from the University of Pittsburgh Medical Center.

A cross-sectional retrospective study was conducted between 2018 and 2021, including 265,554 patients aged between 18 and 26, who received the vaccine either from their primary care physician, pediatrician, or upon direct request to pharmacies. Patients with inconsistent reporting of gender were excluded in order to ensure data integrity and analytical accuracy. The University of Pittsburgh Medical Center Review Board reviewed and approved this study.

### 2.2. Data and Measurement

The primary outcome of this study was HPV vaccination status, classified as none, initiated, and completed (3 doses). Several factors were investigated, including age, gender, race/ethnicity, type of insurance (private, Medicaid, and others), number of primary care (PCP) visits in the past year, and alcohol and tobacco use. The Area Deprivation Index (ADI) 2020, which measures the level of deprivation in the census block group geographic area with a ranking of 1, indicating the lowest level of “disadvantage” within the nation, and an ADI with a ranking of 100, indicating the highest level of “disadvantage”, was also included. Race/ethnicity was classified as Hispanic/Latino (all races), Non-Hispanic (NH) White, NH Black, NH Asian, NH Native American, other, and refused. Additionally, age was classified at first vaccination with the following categories: a recommended age between 11 and 12 years old; early vaccination, 9 years old; early adherent, 9 to 10 years; catch-up, 13–18 years old; and a catch-up between 18 and 26 years old.

### 2.3. Statistical Analysis

The median (interquartile range (IQR)) for continuous variables and counts, and percentages for categorical variables were calculated. The prevalence of HPV vaccination was assessed in the study population and was stratified using the chi-square test for descriptive statistics. A Poisson regression with robust standard errors was performed to calculate prevalence ratios and examine risk factors associated with vaccine initiation and completion. The statistical analysis was conducted using R software for statistical computing (R version 4.3.1). Statistical significance was defined as *p* < 0.05.

## 3. Results

### 3.1. HPV Vaccination Initiation Is More Common among Black Americans Than White Americans

A total of 104,426 patients, with a mean age of 21.89 ± 2.63 years old, received at least one vaccination dose (Table 1).

Overall, 42.2% (67,034) of all the females in the study population received a dose of the HPV vaccine. Females were 37% more likely (PR: 1.37 (95% CI 1.35–1.38)) to initiate the vaccine than males were. When examining vaccination by race, NH Black Americans were 35% more likely (PR: 1.35 (95% CI 1.33–1.37)) to initiate the vaccine compared to NH White Americans. No difference was found regarding insurance, where 40.7% (85,991) and 40.4% (14,109) of patients covered by private insurance and Medicaid initiated the vaccine, respectively. However, in the adjusted model, patients with Medicaid coverage were 8% less likely (PR: 0.92 (95% CI 0.91–0.94)) to initiate the vaccine than those with private insurance coverage. A PCP visit was important for vaccination, where 49.2% of patients who visited a PCP in the past year had received at least one dose of the vaccine, compared to 33.1% who received a vaccine dose with no PCP visit. When stratified by gender (Table 2), 53.9% (9342) of NH Black females and 43.3% (53,923) of NH White females initiated the vaccine. In contrast, 44.7% (4597) of NH Black males and 36.7% (30,678) of NH White males initiated the vaccine.

### 3.2. Characteristics of Patients by HPV Vaccination Completion

A total of 80,480 patients completed the HPV vaccination (Table 3).

Females who completed the full HPV vaccine accounted for a prevalence of 79.6% (53,369). Females were 10% more likely (PR: 1.10 (95% CI 1.09–1.11)) to complete the vaccination than males were. Among NH Black Americans, 68.5% (n = 9551) completed the entire vaccine course, while NH White Americans had a completion percentage of 78.8% (n = 66,668). NH Black Americans were 11% less likely (PR: 0.89 (95% CI 0.88–0.9)) to complete the entire vaccination course when compared to NH White Americans. Patients with private insurance were more likely to complete the HPV vaccine course (78.1%, n = 67,134) than patients with Medicaid (72.2%, n = 10,184) or another type of insurance (73.1%, n = 3162). When adjusted, patients with Medicaid coverage were less likely (PR: 0.96 (95% CI 0.95–0.98)) to complete the vaccine than those with private insurance coverage. Out of the patients who did not have a PCP visit in the past year, 75.5% (40,631) completed the entire vaccine course. Conversely, among patients who visited a PCP, 78.7% (39,849) completed the vaccine. Gender stratification is shown in Table 4.

Completion rates were lower among NH Black individuals compared to those of NH White individuals. Specifically, 71.4% (6670) of NH Black females and 62.7% (2881) of NH Black males completed the HPV vaccine, whereas 81.4% (43,886) of NH White females and 74.3% (22,782) of White males completed the vaccine. NH Black females and males had similar prevalence ratios of 0.89 (95% CI 0.88–0.91) and 0.87 (95% CI 0.85–0.90), respectively, indicating that they were less likely than their White counterparts to complete the HPV vaccine.

### 3.3. Characteristics of Patients by Age at First Vaccination

A total of 25.1% (13,904) of the females received their vaccine dose during the routine age range of 11–12 years old, compared to 15.4% (5004) of the males (Table 5).

Among females, 55.5% (30,807) received their first vaccine dose between the ages of 13 and 17 years (in the catch-up group), while among males, the prevalence was 67.9% (22,109) in this category. When considering races, NH Black Americans had the highest rate of vaccine initiation at early routine age (5.9% (671)) and routine age (25.5% (2916)). NH White Americans had the highest prevalence (61.8% (44,216)) of vaccination during the catch-up phase (13–17 years old). Of the patients who did not visit a PCP, 25.3% (11,224) received the first vaccine dose in the routine age, while 17.6% (7648) of the patients who visited a PCP were in the routine group. Among patients who received the first dose during the routine group, 27.7% (3211) of patients were covered by Medicaid, the highest insurance type rate in this group. In comparison, private insurance had a prevalence of 20.5% (14,938) in the routine group.

## 4. Discussion

The present study aimed to investigate HPV vaccination among patients aged between 18 and 26 years old with at least one encounter at a large healthcare system and identify sociodemographic factors associated with vaccine initiation and completion. In the current sample of 265,554 patients, only 30.3% (80,480) of participants reported having completed the HPV vaccine, while 9.0% (23,946) reported having initiated the HPV vaccine. These rates are substantially lower than the Healthy People 2030 goal of 80%. Moreover, it was found that males were 37% less likely than females to initiate and 10% less likely to complete the vaccination schedule. Although the Advisory Committee on Immunization Practices (ACIP) recommended routine vaccination for males in 2011, vaccination rates for males remain well below those for females. This highlights the importance of further implementing policies to increase male vaccination rates.

In the United States (US), HPV vaccination coverage has been below the recommended optimal levels. Moreover, HPV coverage rates may differ by region and US state. Southern and Midwestern states have exhibited lower rates of HPV vaccination than other states [11,12]. The exact reasons behind geographic disparities in vaccination rates are poorly understood. It seems unlikely that affordability issues are the only factors accounting for lower uptake of HPV vaccines between regions, as the Vaccines for Children program ensures coverage of vaccination costs. In addition, rural populations lag behind urban populations in both vaccine initiation and completion [13,14]. Data from the 2020 National Immunization Survey-Teen (NIS-Teen) showed that vaccination coverage was lower among adolescents living in non-metropolitan statistical areas (MSAs) compared to that of those living in MSA cities [15]. Different barriers to HPV vaccine uptake among rural residents could include poor awareness regarding cancer risks and prevention, significant stigma about sexually transmitted infections, fewer health insurance options, and limited access to healthcare providers and preventive care [16,17,18].

Several studies have demonstrated that racial and ethnic minority populations are less likely to receive vaccine recommendations from healthcare providers [19,20,21,22,23,24]. However, it was found that Black Americans were more likely to initiate the HPV vaccine than White Americans and other race and ethnicities. A possible explanation is that healthcare providers in this region have implemented targeted interventions to increase HPV vaccination rates in minority communities [25,26]. In an urban family medicine residency practice in Pittsburgh, Pennsylvania, a project was conducted from November 2015 to October 2016 to enhance HPV vaccination rates among 9–26-year-old patients from low-income minority populations. Strategies employed included integrating HPV vaccination promotion with community center events (community center mash-ups), sensory-based incentives for interactive vaccine promotion (for example petting a pup or striking a gong after receiving the vaccine), provider training programs, seasonal changing posters, and standing orders for non-physician healthcare professionals to administer the vaccine. As a result, there was a significant 11.3% increase in HPV vaccine initiation and a 12.5% increase in completion rates. [26]. Nevertheless, it was found that despite higher initiation rates among Black Americans, they exhibited lower completion rates. This result aligns with a previous meta-analysis indicating that ethnic/racial minorities tend to have higher vaccine initiation rates but lower completion rates than their White counterparts [27]. This discrepancy might be explained by factors such as a lack of informed discussion about HPV by providers, lack of a regular site of care, less continuity with the same provider, and vaccine hesitancy [27,28,29,30,31,32].

Vaccine hesitancy (i.e., a delay in acceptance or the refusal of vaccination despite the availability of vaccination services) is a crucial barrier to HPV vaccination. In a recent study conducted by Szilagyi et al. [31], it was found that around 23% of parents in the United States with adolescents aged 11–17 displayed hesitancy towards the HPV vaccine, and this hesitancy was strongly linked to a lower likelihood of their adolescents receiving the vaccine [31]. However, it is worth noting that vaccine hesitancy alone does not fully explain the phenomenon of non-vaccination, as the study also revealed that up to 20% of hesitant parents reported that their adolescents still received at least one dose of the HPV vaccine. Moreover, the exact relationship between vaccine hesitancy and sociodemographic factors must be clarified. Indeed, some authors found that the likelihood of delaying/refusing the HPV vaccine was higher among White Americans with higher incomes [33]. In contrast, others in high-income Hispanic communities were characterized by lower hesitancy [31].

Socioeconomic status and health insurance coverage play an important role in vaccination rates. The cost of vaccination is widely recognized as a significant obstacle to initiating and completing the HPV vaccination series in the US [34]. Indeed, a systematic review by Rambout and colleagues found that cost is the primary factor hindering access to the vaccine [35]. Interestingly, most patients in this area presented a relatively high deprivation rate (ADI with a median of 69). However, a previous meta-analysis suggested that the absence of health insurance is even more critical than income in predicting non-vaccination, which has significant implications for vaccination strategies [36]. Accordingly, an association was found between insurance type and HPV vaccination status in this study. The findings showed that private insurance patients were less likely to initiate HPV vaccination than patients with other insurance types. However, patients with private insurance were found to be more prone to complete HPV vaccination than those with either Medicaid or another insurance type. Interestingly, Sriram et al. also found that adolescents with private health insurance coverage displayed a lower likelihood of receiving the HPV vaccine compared to that of those covered by Medicaid health insurance plans [37]. Conversely, data from the 2019 NIS-Teen showed that adolescents with Medicaid were more likely to complete HPV vaccination than those with private insurance or without insurance [38]. One possible explanation for this disparity could be the Medicaid expansion facilitated by the Patient Protection and Affordable Care Act (ACA) in 2014, which provided newly eligible enrollees with access to HPV vaccine coverage [39]. However, the exact reasons underlying the observed differences in HPV vaccine uptake across various insurance coverages remain uncertain. Thus, further investigation is necessary to explore the potential impact of restrictions imposed by private insurance companies on the acceptance of HPV vaccines.

According to a study by Berdnarczyk et al. [40], approximately 4% of US adolescents initiated the HPV vaccine between the ages of 9 and 10, whereas 51% began receiving the vaccine between the ages of 11 and 12. Moreover, although most of those initiating vaccination before age 14 eventually completed the series (>85% by age 17), only 20% of those starting the series at 15 or older achieved series completion. This study showed that 1.5% of males and 5.6% of females received their first HPV vaccine dose between 9 and 10 years old, while most patients received their first dose between 13 and 17 years old (55.5% of females and 67.9% of males). It was also found that despite the recommended age for vaccination initiation being 11–12 years old, the vaccine initiation rate was only 25.1% for females and 15.4% for males within that specific age group. This highlights the urgent need to collaborate and raise awareness about the importance of HPV vaccination and the recommended age for initiation by launching an educational campaign targeting healthcare providers, parents, and schools.

Finally, it has been demonstrated that access to a primary care provider is a crucial factor associated with improved HPV vaccination rates [41], which was also seen in this study. There is a need for further interventions to assist providers in recommending HPV vaccination effectively and efficiently. These interventions should account for factors that might influence HPV vaccine uptake, both provider- and patient-related. Provider-related factors might include time constraints, knowledge of the vaccine, self-efficacy in discussing the vaccine, and confidence in its efficacy. On the other hand, patient-related factors might include trust issues, perceived high cost, perceived side effects, and concerns about adolescent sexual activity [42]. From this perspective, it is imperative to implement education and training programs to overcome these existing limitations.

Despite its strengths and significant research findings, this study has some limitations. First, as this is a cross-sectional descriptive study, causality cannot be inferred from the observed associations. Second, the study focuses on patients aged 18–26 years with at least one encounter at a large healthcare system. Thus, the observed trends may not be generalizable and fully represent the wider population. Finally, HPV status was abstracted from the clinical intake forms, which is subject to potential misclassification by recall and may have introduced bias that can impact validity accuracy. However, the subjects were also randomly sampled for clinical chart review, and high concurrence was found.

## 5. Conclusions

In conclusion, HPV vaccination rates among patients aged 18–26 years within a large healthcare system were significantly below national rates and far from the Healthy People 2030 goal. Implementing targeted education and training programs becomes imperative, aiming to provide individuals, particularly males, with the necessary knowledge to make informed decisions regarding vaccination. Furthermore, enhancing healthcare providers’ communication skills by effectively conveying the benefits and safety of vaccines can effectively alleviate concerns and uncertainties surrounding vaccination. Moreover, efforts should be made to ensure that education and training programs are accessible, inclusive, and tailored to the diverse needs of different communities. Engaging with individuals and communities, gaining insights into their specific concerns, and adapting communication strategies, accordingly, will help build a more trusting and supportive environment that promotes vaccine acceptance. Therefore, empowering individuals through knowledge, dispelling myths, and nurturing trust can significantly increase vaccination rates.

## Figures and Tables

**Table 1 cancers-15-04243-t001:** Characteristics of patients aged 18–26 with at least one UPMC encounter between 2018 and 2021 by HPV vaccination initiation status.

	No Vaccination	Vaccination Initiation			
	*n* = 161,128	*n* = 104,426			
	*n* (%)	*n* (%)	*p*	RR (95% CI)	*p*
Gender					
Male	69,134 (64.9)	37,392 (35.1)	<0.001	1	
Female	91,994 (57.8)	67,034 (42.2)		1.37 (1.35, 1.38)	<0.001
Race/Ethnicity					
White	123,569 (59.4)	84,601 (40.6)	<0.001	1	<0.001
Black	13,663 (49.5)	13,939 (50.5)		1.35 (1.33, 1.37)	<0.001
American Indian	403 (66.8)	200 (33.2)		0.89 (0.79, 1.01)	<0.001
AAPI	5157 (73.1)	1902 (26.9)		0.69 (0.66, 0.72)	<0.001
Hispanic	2779 (66.1)	1425 (33.9)		0.87 (0.83, 0.91)	<0.001
Other	506 (86.5)	79 (13.5)		0.42 (0.34, 0.52)	<0.001
Declined	3608 (78.7)	976 (21.3)		0.56 (0.52, 0.59)	<0.001
Missing	11,443	1304		--	
PCP visit					
No	108,824 (66.9)	53,796 (33.1)	<0.001	1	
Yes	52,304 (50.8)	50,630 (49.2)		1.81 (1.79, 1.82)	<0.001
Insurance type					
Private	125,176 (59.3)	85,991 (40.7)	<0.001	1	
Medicaid	20,804 (59.6)	14,109 (40.4)		0.92 (0.91, 0.94)	<0.001
Other	15,148 (77.8)	4326 (22.2)		0.65 (0.63, 0.67)	<0.001
Tobacco					
No	130,962 (59.7)	88,573 (40.3)	0.787	1	
Yes	22,345 (59.7)	15,065 (40.3)		1.04 (1.03, 1.06)	<0.001
Missing	7821	788		--	
Alcohol					
No	78,761 (60.4)	51,605 (39.6)	<0.001	1	
Yes	54,184 (63.5)	31,087 (36.5)		0.96 (0.95, 0.97)	<0.001
Missing	28,183	21,734		--	
Illicit drugs					
No	112,517 (61.1)	71,575 (38.9)	<0.001	--	--
Yes	11,003 (58.3)	7876 (41.7)		--	
Missing	37,608	24,975		--	
Age (mean (SD))	22.57 (2.50)	21.89 (2.63)	<0.001	0.94 (0.94, 0.94)	<0.001
ADI					
[75, 100]	56,433 (62.2)	34,301 (37.8)	<0.001	1	
[1, 25)	5805 (84.3)	1080 (15.7)		0.56 (0.52, 0.59)	<0.001
[25, 50)	26,390 (53.8)	22,637 (46.2)		1.33 (1.31, 1.35)	<0.001
[50, 75)	71,894 (60.8)	46,342 (39.2)		1.09 (1.08, 1.1)	<0.001

*p*: *p*-value; RR: risk ratio; CI: confidence interval; AAPI: Asian American Pacific Islander; PCP: Primary Care Physician; SD: standard deviation; ADI: Area Deprivation Index.

**Table 2 cancers-15-04243-t002:** HPV vaccination initiation among patients aged 18–26 with at least one UPMC encounter between 2018 and 2021 by gender.

	Females					Males				
	No Vaccination	Vaccination Initiation				No Vaccination	Vaccination Initiation			
	*n* = 91,994	*n* = 67,034				*n* = 69,134	*n* = 37,392			
	*n* (%)	*n* (%)	*p*	RR (95% CI)	*p*	*n* (%)	*n* (%)	*p*	RR (95% CI)	*p*
Race/Ethnicity										
White	70,674 (56.7)	53,923 (43.3)	<0.001	1		52,895 (63.3)	30,678 (36.7)	<0.001	1	
Black	7976 (46.1)	9342 (53.9)		1.34 (1.32, 1.37)	0.008	5687 (55.3)	4597 (44.7)		1.35 (1.31, 1.39)	<0.001
American Indian	239 (66.2)	122 (33.8)		0.81 (0.70, 0.94)	0.076	164 (67.8)	78 (32.2)		1.08 (0.90, 1.31)	0.403
AAPI	2892 (69.5)	1272 (30.5)		0.71 (0.68, 0.75)	0.024	2265 (78.2)	630 (21.8)		0.67 (0.62, 0.72)	0.000
Hispanic	1644 (63.7)	935 (36.3)		0.86 (0.82, 0.91)	0.027	1135 (69.8)	490 (30.2)		0.89 (0.82, 0.97)	0.009
Other	264 (84.6)	48 (15.4)		0.4 (0.31, 0.53)	0.140	242 (88.6)	31 (11.4)		0.45 (0.32, 0.63)	<0.001
Declined	1986 (77.1)	589 (22.9)		0.56 (0.52, 0.61)	0.040	1622 (80.7)	387 (19.3)		0.56 (0.50, 0.62)	<0.001
Missing	6319	803		--		5124	501		--	
PCP visit										
No	65,818 (66.3)	33,415 (33.7)	<0.001	1		43,006 (67.8)	20,381 (32.2)	<0.001	1	
Yes	26,176 (43.8)	33,619 (56.2)		1.84 (1.82, 1.86)	0.006	26,128 (60.6)	17,011 (39.4)		1.75 (1.72, 1.79)	<0.001
Insurance type				--					--	
Private	71,079 (56.5)	54,776 (43.5)	<0.001	1		54,097 (63.4)	31,215 (36.6)	<0.001	1	
Medicaid	13,229 (58.2)	9485 (41.8)		0.92 (0.90, 0.94)	0.009	7575 (62.1)	4624 (37.9)		0.94 (0.91, 0.97)	<0.001
Other	7686 (73.5)	2773 (26.5)		0.69 (0.66, 0.71)	0.017	7462 (82.8)	1553 (17.2)		0.59 (0.56, 0.62)	<0.001
Tobacco										
No	77,631 (57.3)	57,805 (42.7)	<0.001	1		53,331 (63.4)	30,768 (36.6)	<0.001	1	
Yes	10,313 (53.9)	8809 (46.1)		1.08 (1.06, 1.1)	0.009	12,032 (65.8)	6256 (34.2)		1.01 (0.99, 1.04)	<0.001
Missing	4050	420		--		3771	368		--	
Alcohol										
No	48,166 (58.4)	34,241 (41.6)	<0.001	1		30,595 (63.8)	17,364 (36.2)	<0.001	1	
Yes	30,251 (57.5)	22,377 (42.5)		1.00 (0.99, 1.01)	0.007	23,933 (73.3)	8710 (26.7)		0.91 (0.89, 0.93)	0.349
Missing	13,577	10,416		--		14,606	11,318			
Illicit drugs										
No	68,318 (57.7)	50,095 (42.3)	<0.001	--	--	44,199 (67.3)	21,480 (32.7)	<0.001	--	--
Yes	5768 (54.3)	4859 (45.7)		--		5235 (63.4)	3017 (36.6)		--	
Missing	17,908	12,080		--		19,700	12,895		--	
Age (mean (SD))	22.63 (2.49)	22.33 (2.64)	<0.001	0.97 (0.97, 0.97)	0.001	22.48 (2.51)	21.11 (2.42)	<0.001	0.89 (0.88, 0.89)	<0.001
ADI										
[75, 100]	33,067 (59)	22,944 (41)	<0.001	1		23,366 (67.3)	11,357 (32.7)	<0.001	1	
[1, 25)	3401 (83.4)	678 (16.6)		0.55 (0.51, 0.59)	0.039	2404 (85.7)	402 (14.3)		0.58 (0.52, 0.65)	<0.001
[25, 50)	14,114 (51.2)	13,457 (48.8)		1.29 (1.26, 1.31)	0.009	12,276 (57.2)	9180 (42.8)		1.43 (1.4, 1.47)	<0.001
[50, 75)	41,077 (57.9)	29,908 (42.1)		1.07 (1.06, 1.09)	0.007	30,817 (65.2)	16,434 (34.8)		1.12 (1.1, 1.15)	<0.001

*p*: *p*-value; RR: risk ratio; CI: confidence interval; AAPI: Asian American Pacific Islander; PCP: Primary Care Physician; SD: standard deviation; ADI: Area Deprivation Index.

**Table 3 cancers-15-04243-t003:** Characteristics of patients aged 18–26 with at least one UPMC encounter between 2018 and 2021 by HPV vaccination status.

	Vaccination Initiation	Vaccination Completion			
	*n* = 23,946	*n* = 80,480			
	*n* (%)	*n* (%)	*p*	RR (95% CI)	*p*
Gender					
Male	10,281 (27.5)	27,111 (72.5)		1	
Female	13,665 (20.4)	53,369 (79.6)		1.10 (1.09, 1.11)	<0.001
Race/Ethnicity					
White	17,933 (21.2)	66,668 (78.8)	<0.001	1.00	
Black	4388 (31.5)	9551 (68.5)		0.89 (0.88, 0.9)	0.000
American Indian	46 (23)	154 (77)		0.97 (0.89, 1.05)	0.440
AAPI	486 (25.6)	1416 (74.4)		0.93 (0.9, 0.96)	0.000
Hispanic	367 (25.8)	1058 (74.2)		0.94 (0.91, 0.97)	0.000
Other	30 (38)	49 (62)		0.82 (0.69, 0.97)	0.025
Declined	271 (27.8)	705 (72.2)		0.92 (0.88, 0.96)	0.000
Missing	425	879		--	
PCP visit					
No	13,165 (24.5)	40,631 (75.5)	<0.001	1	
Yes	10,781 (21.3)	39,849 (78.7)		1.03 (1.02, 1.04)	<0.001
Insurance type					
Private	18,857 (21.9)	67,134 (78.1)	<0.001	1	
Medicaid	3925 (27.8)	10,184 (72.2)		0.96 (0.95, 0.98)	<0.001
Other	1164 (26.9)	3162 (73.1)		0.95 (0.93, 0.97)	<0.001
Tobacco					
No	19,582 (22.1)	68,991 (77.9)	<0.001	1	
Yes	4090 (27.1)	10,975 (72.9)		0.95 (0.94, 0.96)	0.000
Missing	274	514		--	
Alcohol					
No	12,381 (24)	39,224 (76)	<0.001		
Yes	5975 (19.2)	25,112 (80.8)		1.01 (1, 1.02)	0.010
Missing	5590	16,144		--	
Illicit drugs					
No	15,480 (21.6)	56,095 (78.4)	<0.001	--	
Yes	2143 (27.2)	5733 (72.8)		--	--
Missing	6323	18,652		--	
Age (mean (SD))	21.41 (2.76)	22.03 (2.57)		1.01 (1.01, 1.02)	<0.001
ADI					
[75, 100]	8845 (25.8)	25,456 (74.2)	<0.001	1	
[1, 25)	243 (22.5)	837 (77.5)		1.04 (1.00, 1.08)	0.025
[25, 50)	4692 (20.7)	17,945 (79.3)		1.05 (1.04, 1.06)	0.000
[50, 75)	10,144 (21.9)	36,198 (78.1)		1.03 (1.02, 1.04)	0.000
Age at vaccination (mean (SD))	16.21 (3.67)	13.97 (2.75)	<0.001	--	

*p*: *p*-value; RR: risk ratio; CI: confidence interval; AAPI: Asian American Pacific Islander; PCP: Primary Care Physician; SD: standard deviation; ADI: Area Deprivation Index.

**Table 4 cancers-15-04243-t004:** HPV vaccination completion among patients aged 18–26 with at least one UPMC encounter between 2018 and 2021 by gender.

	Females					Males				
	Vaccination Initiation	Vaccination Completion				Vaccination Initiation	Vaccination Completion			
	*n* = 13,665	*n* = 53,369				*n* = 10,281	*n* = 27,111			
	*n* (%)	*n* (%)	*p*	RR (95% CI)	*p*	*n* (%)	*n* (%)	*p*	RR (95% CI)	*p*
Race/Ethnicity										
White	10,037 (18.6)	43,886 (81.4)	<0.001	1		7896 (25.7)	22,782 (74.3)	<0.001	1	
Black	2672 (28.6)	6670 (71.4)		0.89 (0.88, 0.91)	<0.001	1716 (37.3)	2881 (62.7)		0.87 (0.85, 0.90)	<0.001
American Indian	25 (20.5)	97 (79.5)		0.97 (0.88, 1.07)	0.515	21 (26.9)	57 (73.1)		0.96 (0.82, 1.14)	0.662
AAPI	285 (22.4)	987 (77.6)		0.94 (0.91, 0.97)	<0.001	201 (31.9)	429 (68.1)		0.90 (0.85, 0.96)	0.002
Hispanic	233 (24.9)	702 (75.1)		0.92 (0.89, 0.96)	<0.001	134 (27.3)	356 (72.7)		0.97 (0.91, 1.04)	0.393
Other	17 (35.4)	31 (64.6)		0.77 (0.61, 0.97)	0.026	13 (41.9)	18 (58.1)		0.91 (0.70, 1.18)	0.466
Declined	150 (25.5)	439 (74.5)		0.91 (0.86, 0.96)	0.001	121 (31.3)	266 (68.7)		0.95 (0.87, 1.03)	0.190
Missing	246	557		--		179	322		--	
PCP visit										
No	7570 (22.7)	25,845 (77.3)	<0.001	1		5595 (27.5)	14,786 (72.5)	<0.001	1	
Yes	6095 (18.1)	27,524 (81.9)		1.03 (1.02, 1.04)	<0.001	4686 (27.5)	12,325 (72.5)		1.02 (1.01, 1.04)	0.007
Insurance type										
Private	10,615 (19.4)	44,161 (80.6)	<0.001	1		8242 (26.4)	22,973 (73.6)	<0.001	1	
Medicaid	2420 (25.5)	7065 (74.5)		0.96 (0.95, 0.98)	<0.001	1505 (32.5)	3119 (67.5)		0.95 (0.92, 0.98)	<0.001
Other	630 (22.7)	2143 (77.3)		0.97 (0.95, 0.99)	0.003	534 (34.4)	1019 (65.6)		0.91 (0.88, 0.95)	<0.001
Tobacco										
No	11,389 (19.7)	46,416 (80.3)		1		8193 (26.6)	22,575 (73.4)		1	
Yes	2136 (24.2)	6673 (75.8)		0.95 (0.94, 0.97)	<0.001	1954 (31.2)	4302 (68.8)		0.95 (0.92, 0.97)	<0.001
Missing	140	280		--		134	234		--	
Alcohol										
No	7464 (21.8)	26,777 (78.2)	<0.001	1		4917 (28.3)	12,447 (71.7)	<0.001	1	
Yes	3643 (16.3)	18,734 (83.7)		1.02 (1.01, 1.03)	<0.001	2332 (26.8)	6378 (73.2)		1.00 (0.98, 1.01)	0.629
Missing	2558	7858		--		3032	8286		--	
Illicit drugs										
No	9641 (19.2)	40,454 (80.8)	<0.001	--	--	5839 (27.2)	15,641 (72.8)	<0.001	--	--
Yes	1167 (24)	3692 (76)		--		976 (32.4)	2041 (67.6)		--	
Missing	2857	9223		--		3466	9429		--	
Age (mean (SD))	21.72 (2.82)	22.48 (2.56)	<0.001	1.02 (1.02, 1.02)	<0.001	21.00 (2.63)	21.15 (2.34)	<0.001	1.01 (1, 1.01)	0.002
ADI										
[75, 100]	5268 (23)	17,676 (77)	<0.001			3577 (31.5)	7780 (68.5)	<0.001		
[1, 25)	140 (20.6)	538 (79.4)		1.03 (0.99, 1.07)	0.147	103 (25.6)	299 (74.4)		1.07 (1, 1.15)	0.051
[25, 50)	2438 (18.1)	11,019 (81.9)		1.04 (1.02, 1.05)	<0.001	2254 (24.6)	6926 (75.4)		1.08 (1.05, 1.1)	<0.001
[50, 75)	5808 (19.4)	24,100 (80.6)		1.03 (1.02, 1.04)	<0.001	4336 (26.4)	12,098 (73.6)		1.04 (1.02, 1.06)	<0.001
Age at Vaccination (mean (SD))	16.18 (3.96)	13.69 (2.79)	<0.001	--	--	16.25 (3.24)	14.50 (2.61)	<0.001	--	--

*p*: *p*-value; RR: risk ratio; CI: confidence interval; AAPI: Asian American Pacific Islander; PCP: Primary Care Physician; SD: standard deviation; ADI: Area Deprivation Index.

**Table 5 cancers-15-04243-t005:** Characteristics of patients aged 18–26 with at least one UPMC encounter between 2018 and 2021 by age at first vaccination.

	Early: <9 Years	Early Routine: 9–10 Years	Routine: 11–12 Years	Catch Up: 13–17 Years	Catch Up: 18–26 Years	*p*-Value
	*n* = 285	*n* = 3615	*n* = 18,843	*n* = 52,916	*n* = 12,338	
	*n* (%)	*n* (%)	*n* (%)	*n* (%)	*n* (%)	
Gender						
Male	51 (0.2)	498 (1.5)	5004 (15.4)	22,109 (67.9)	4915 (15.1)	<0.001
Female	234 (0.4)	3117 (5.6)	13,904 (25.1)	30,807 (55.5)	7423 (13.4)	
Race/Ethnicity						
White	229 (0.3)	2778 (3.9)	15,138 (21.2)	44,216 (61.8)	9176 (12.8)	<0.001
Black	40 (0.4)	671 (5.9)	2916 (25.5)	6063 (53.1)	1734 (15.2)	
American Indian	1 (0.6)	9 (5.7)	34 (21.5)	89 (56.3)	25 (15.8)	
AAPI	0 (0)	40 (2.4)	217 (12.8)	855 (50.5)	580 (34.3)	
Hispanic	6 (0.5)	65 (5.4)	263 (22)	626 (52.3)	236 (19.7)	
Other	0 (0)	0 (0)	12 (16.7)	40 (55.6)	20 (27.8)	
Declined	4 (0.5)	27 (3.2)	161 (19.1)	463 (55.1)	186 (22.1)	
Missing	167	5	0	25	564	
PCP visit						
No	125 (0.3)	1809 (4.1)	11,224 (25.3)	26,906 (60.6)	4367 (9.8)	<0.001
Yes	160 (0.4)	1806 (4.1)	7684 (17.6)	26,010 (59.6)	7971 (18.3)	
Insurance type						
Private	216 (0.3)	2736 (3.8)	14,938 (20.5)	44,422 (61.1)	10,432 (14.3)	<0.001
Medicaid	46 (0.4)	745 (6.4)	3211 (27.7)	6277 (54.1)	1322 (11.4)	
Other	23 (0.6)	134 (3.6)	759 (20.4)	2217 (59.6)	584 (15.7)	
Tobacco						
No	245 (0.3)	2983 (4)	15,828 (21.2)	45,023 (60.3)	10,631 (14.2)	<0.001
Yes	39 (0.3)	613 (4.8)	2941 (23.2)	7500 (59.1)	1593 (12.6)	
Missing	1	19	139	393	114	
Alcohol						
No	134 (0.3)	2125 (4.9)	10,387 (24.1)	25,241 (58.7)	5131 (11.9)	<0.001
Yes	117 (0.4)	882 (3.2)	3819 (14)	16613 (61)	5783 (21.3)	
Missing	34	608	4688	11,062	1424	
Illicit drugs						
No	223 (0.4)	2605 (4.3)	12,357 (20.3)	36,144 (59.5)	9444 (15.5)	0.253
Yes	24 (0.4)	310 (4.6)	1338 (19.9)	3943 (58.7)	1100 (16.4)	
Missing	38	700	5200	12,829	1794	
Vaccination frequency						
Initiation	25 (0.1)	237 (1.1)	1957 (9.1)	12,720 (59.2)	6551 (30.5)	<0.001
Completion	260 (0.4)	3378 (5.1)	16,951 (25.5)	40,196 (60.4)	5787 (8.7)	
Age (mean (SD))	22.67 (2.49)	21.19 (2.17)	20.81 (2.32)	22.10 (2.64)	23.67 (2.21)	<0.001
ADI						
[75, 100]	121 (0.4)	1451 (5.1)	6998 (24.5)	16,300 (57)	3740 (13.1)	<0.001
[1, 25)	2 (0.2)	18 (1.9)	138 (14.9)	585 (63)	186 (20)	
[25, 50)	45 (0.2)	491 (2.5)	2967 (15.3)	12,954 (66.9)	2904 (15)	
[50, 75)	117 (0.3)	1654 (4.2)	8794 (22.5)	23,039 (58.9)	5498 (14.1)	

*p*: *p*-value; AAPI: Asian American Pacific Islander; PCP: Primary Care Physician; SD: standard deviation; ADI: Area Deprivation Index.

## Data Availability

The data presented in this study are available on request from the corresponding author.

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
