# Peer review of "Barriers to Human Papillomavirus Vaccination Initiation and Completion among Adults Aged 18–26 Years in a Large Healthcare System"

_cancers, 2023, doi:10.3390/cancers15174243_

Round 1

Reviewer 1 Report

Publishable with minor changes
This is a well articulated work assessing the factors for completion of a HPV vaccine series in 265,554 patients aged 18-26 in a large health care system. It is well written. There are some comments I have related to readers better understanding the context of the work.
1. What does "large healthcare system" mean? Is this a state or private or medicaide/medicare run system? It would be important especially for readers from another country to understand the context.
2. This work asks a question that involved the COVID era (2020 forward). This work addresses 2018-2021. No where in the analysis or the discussion do the authors illude to the impact COVID had on the results.
3. Pg 3 Line190-191 The authors use the phrase "access to". Is this the correct phrase or is it more related to "uptake of"?
4. Another context question that could help the reader - during this time frame, where to individuals get the vaccine - School based programs, family doctors offices, pediatrician offices, direct request to pharmacies, Public health clinic?
5. Pg 3. Line 207-208. Explanatory question - please provide a definition or reference or example of  "mash-ups", "sensory based incentives" ect
6. Pg 4 - Another context question - was the HPV vaccine free of charge or did it cost the individual or was there a service fee? 

Publishable with minor changes
This is a well articulated work assessing the factors for completion of a HPV vaccine series in 265,554 patients aged 18-26 in a large health care system. It is well written. There are some comments I have related to readers better understanding the context of the work.

Author Response

  • To clarify the meaning of “large healthcare system”.

We thank the Reviewer for this comment. We completely agree that the expression “large healthcare system” might lack clarity, especially for readers from other countries. In order to enhance clarity, we explicitly specified in the “Introduction” section (page 2) that our study was conducted across the University of Pittsburgh Medical Center (UPMC) hospital system in Pennsylvania, revising the passage as follows: “In our study, we provide an estimate of HPV vaccination uptake among eligible adults aged between 18 and 26 years old within the University of Pittsburgh Medical Center (UPMC) hospital system, which is a large private healthcare system operating multiple hospitals. Our aim is to identify which sociodemographic factors are associated with low vaccination initiation and completion in this population to reduce the burden of diseases related to HPV in vulnerable populations.”

  • To discuss the impact of COVID-19 on our results.

We greatly appreciate the insights provided by the Reviewer. We acknowledge the potential influence of the COVID-19 pandemic on our observed results. To address this, we are committed to conducting a distinct upcoming study where we will comprehensively examine both COVID-19 vaccination rates and HPV vaccination rates within the same sample population. Our aim is to assess and compare the prevalence of COVID-19 vaccination in relation to HPV vaccination within our study population. We plan to examine the vaccination rates for each vaccine according to age at vaccination, in order to identify potential differences in vaccine uptake across different age groups. Additionally, we want to investigate the potential repercussions of the COVID-19 pandemic on HPV vaccination rates. Our analysis will seek to ascertain whether the pandemic prompted heightened hesitancy and consequently a reduced uptake of the HPV vaccine in contrast to pre-pandemic era. Conversely, we will explore the possibility of HPV vaccination rates remaining steady despite the disruptive influence of the COVID-19 pandemic.

  • To check the accuracy of the phrase “access to” in the second paragraph of the “Discussion” section.

We completely agree with the Reviewer that that the phrase "uptake of" is a more fitting choice than "access to" within this particular context. The use of "uptake" precisely conveys our interest in the active acceptance of vaccines by individuals, irrespective of the vaccines’ availability. In contrast, "access to" might erroneously imply an unavailability of vaccines, which is not our intended message in this context. Accordingly, we replaced "access to" with "uptake of" in the second paragraph of the “Discussion” section.

  • To specify where the included patients received the vaccine.

We totally understand this Reviewer’s comment because we did not specify where individuals received the vaccine. Accordingly, we revised the “Study design” paragraph of the “Materials and Methods” section (page 2) as follows: “We conducted a cross-sectional retrospective study between 2018 and 2021, including 265,554 patients aged between 18 and 26, who received the vaccine either from their primary care physician, pediatrician or upon direct request to pharmacies.”

  • To better clarify the meaning of “mash-ups", "sensory based incentives", etc.

We thank the Reviewer for this comment. We acknowledge the potential unfamiliarity of many readers with the definitions of the abovementioned interventions. Accordingly, in order to better define these concepts, we revised the third paragraph of the “Discussion” section as follows: “Strategies employed included integrating HPV vaccination promotion with community center events (community center mash-ups), sensory-based incentives for interactive vaccine promotion (for example petting a pup or striking a gong after receiving the vaccine), provider training programs, seasonal changing posters, and standing orders for non-physician healthcare professionals to administer the vaccine.”

  • To specify whether the HPV vaccine was provided at no cost, incurred an individual expense, or involved a service fee.

  • HPV vaccine incurred an individual expense.

Reviewer 2 Report

Dear authors,

RE:Barriers to Human Papillomavirus vaccination initiation and completion among adults aged 18-26 years in a large healthcare system”by Lucien Khalil et al

Despite the limitations and possible mis-perfections of the study, my impression is that the particular well-conducted study should be considered for publication in your prestigious journal.

Additionally, similar publications by other scientific groups in the particular field, especially in countries that have not yet included anti-HPV vaccine in their national vaccination schedule, should also be encouraged in terms of reaching the WHO goals.

I do have some comments for the authors:

Please consider adding some sentences or a paragraph presenting additional potential benefits from vaccination; explaining the effectiveness of the vaccine and the possible positive effect in already HPV positive women in terms of clearance of the infection/disease.

You can use the following references:

1.     Paraskevaidis E, Athanasiou A, Paraskevaidi M, Bilirakis E, Galazios G, et. al.  Cervical Pathology Following HPV Vaccination in Greece: A 10-year HeCPA Observational Cohort Study. In Vivo. 2020 May-Jun;34(3):1445-1449. doi: 10.21873/invivo.11927. PMID: 32354944; PMCID: PMC7279786.

2.     Alterations of HPV-Related Biomarkers after Prophylactic HPV Vaccination. A Prospective Pilot Observational Study in Greek Women.

Valasoulis G, Pouliakis A, Michail G, Kottaridi C, Spathis A, Kyrgiou M, Paraskevaidis E, Daponte A. Cancers (Basel). 2020 May 5;12(5):1164.

doi: 10.3390/cancers12051164. PMID: 32380733; PMCID: PMC7281708.

Author Response

  • To discuss additional potential advantages of vaccination, elucidating the vaccine's efficacy and the potential favorable impact on women who are already HPV-positive, particularly concerning the clearance of the infection or disease.

We totally agree with the Reviewer that we need to discuss additional potential benefits of HPV vaccination. HPV vaccination can provide strong protection against cervical cancer and other HPV-related diseases. Indeed, the HPV vaccine has been shown to be extremely effective, reducing the risk of persistent HPV infection in vaccinated individuals. It can markedly reduce development of severe cervical precancers and consequently the need for treatment, as well as their long-term related obstetrical morbidity. HPV vaccination was also shown to reduce the prevalence of HPV 16, 18 and 31 infections among those who tested positive for these high-risk HPV genotypes. Accordingly, we added the following sentence in the “introduction” section (page 2): “HPV vaccination can provide benefits even in cases of existing infection. Remarkably, it has shown the capability to diminish the risk of persistent HPV infection and aid in the regression of cervical precancerous lesions. Furthermore, it appears to play a role in decreasing the prevalence of high-risk HPV genotype infections among individuals who have previously tested positive for them.” We also added some specific references, as suggested.

Reviewer 3 Report

Dear authors,

very interesting paper. It is well-written and with large population. However I think you should expand some points:

Line 48: I would also expand discussion on the low knowledge of second primary HPV related tumors (e.g. Preti M, Rosso S, Micheletti L, Libero C, Sobrato I, Giordano L, Busso P, Gallio N, Cosma S, Bevilacqua F, Benedetto C. Risk of HPV-related extra-cervical cancers in women treated for cervical intraepithelial neoplasia. BMC Cancer. 2020 Oct 7;20(1):972. doi: 10.1186/s12885-020-07452-6. PMID: 33028248; PMCID: PMC7542855.)

Line 217: also expand here discussion abpur second primary cancers. Can it be a further stimulus for HPV vaccination?

Thank you for your precious work.

Author Response

  • To discuss the limited awareness regarding second primary tumors associated with HPV.

We thank the Reviewer for this comment. We completely agree that we need to discuss the limited awareness regarding second primary tumors associated with HPV. Indeed, there is a gap in knowledge regarding the risk of developing secondary cancers associated with HPV infection. Increased awareness and improved screening protocols are needed to better detect potential second primary tumors in those already diagnosed with an initial HPV-related cancer. Accordingly, we added the following sentence in the “Introduction” section (page 2): “Moreover, it has been suggested that women who have undergone prior treatment for cervical precancerous lesions could face an elevated risk of developing subsequent HPV-associated malignancies. Nonetheless, there is limited understanding regarding the extent of risk associated with these secondary tumors, as well as the efficacy of HPV vaccination in individuals who have already received treatment.” We also added the suggested reference.

  • To further discuss the risk of second primary tumors associated with HPV in the “Discussion” section.

While we greatly value the comment suggesting a discussion of the risk of second primary tumors associated with HPV in the "Discussion" section, we opted not to address this aspect in our study. Our primary focus centered on analyzing the sociodemographic factors influencing HPV vaccination rates among patients aged between 18-26 years old with at least one encounter at a large healthcare system. Regrettably, the inclusion of the second primary tumor risk discussion didn't align with the scope of our analysis. Our aim was to comprehensively examine the impact of sociodemographic variables on vaccination rates, which we believe provides a robust understanding of the factors influencing HPV vaccine uptake.
